# Correspondence Learning via Linearly-invariant Embedding

**Riccardo Marin**[*]
University of Verona
riccardo.marin_01@univr.it

**Marie-Julie Rakotosaona**[*]
LIX, Ecole Polytechnique, IP Paris
mrakotos@lix.polytechnique.fr

**Simone Melzi**
LIX, Ecole Polytechnique, IP Paris
Sapienza University of Rome
melzi@di.uniroma1.it

**Maks Ovsjanikov**
LIX, Ecole Polytechnique, IP Paris
maks@lix.polytechnique.fr

## Abstract

In this paper, we propose a fully differentiable pipeline for estimating accurate dense correspondences between 3D point clouds. The proposed pipeline is an extension and a generalization of the *functional maps framework*. However, instead of using the Laplace-Beltrami eigenfunctions as done in virtually all previous works in this domain, we demonstrate that learning the basis from data can both improve robustness and lead to better accuracy in challenging settings. We interpret the basis as a learned embedding into a higher dimensional space. Following the functional map paradigm the optimal transformation in this embedding space must be linear and we propose a separate architecture aimed at estimating the transformation by learning optimal descriptor functions. This leads to the first end-to-end trainable functional map-based correspondence approach in which both the basis and the descriptors are learned from data. Interestingly, we also observe that learning a *canonical* embedding leads to worse results, suggesting that leaving an extra linear degree of freedom to the embedding network gives it more robustness, thereby also shedding light onto the success of previous methods. Finally, we demonstrate that our approach achieves state-of-the-art results in challenging non-rigid 3D point cloud correspondence applications.

## 1   Introduction

Computing correspondences between geometric objects is a widely investigated task. Its applications are countless: rigid and non-rigid registration methods are instrumental in engineering, medicine and biology [25, 29, 16] among other fields. Point cloud registration is important for range scan data, e.g., in robotics [19, 52], but the problem can also be generalized to abstract domains like graphs [58, 15].

The *non-rigid* correspondence problem is particularly challenging as a successful solution must deal with a large variability in shape deformations and be robust to noise in the input data. To address this problem, in recent years, several data-driven approaches have been proposed to learn the optimal transformation model from data rather than imposing it *a priori*, including [20, 61, 7] among others. In this domain, a prominent direction is based on the functional map representation [39], which has been adapted to the learning-based setting [31, 22, 48, 13]. These methods have shown that optimal feature or descriptor functions (also known as "probe" functions) can be learned from data and then used successfully within the functional map pipeline to obtain accurate dense correspondences.

---

[*]denotes equal contribution.

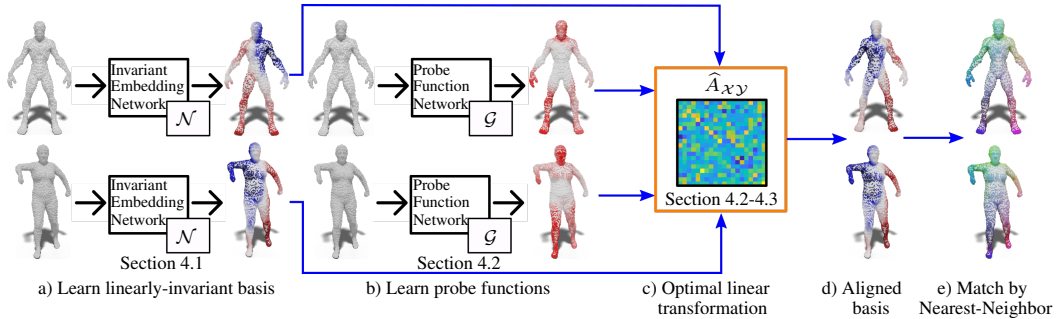

a) Learn linearly-invariant basis    b) Learn probe functions    c) Optimal linear transformation    d) Aligned basis    e) Match by Nearest-Neighbor

Figure 1: Pipeline overview: starting from point cloud coordinates we obtain a set of linearly-invariant basis functions via the Invariant Embedding Network $\mathcal{N}$ (a), and descriptors using the Probe Function Network $\mathcal{G}$ (b). The learned basis and probe functions are used to compute the optimal linear transformation $\widehat{A}_{\mathcal{XY}}$ (c). This transformation is used to align the two sets of bases (d). The correspondence between point clouds is then estimated using nearest neighbors between the aligned basis sets (e). Note that the underlying meshes are depicted only for sake of clarity of visualization.

Unfortunately, the reduced functional basis, which forms the key ingredient in this approach, has so far been tied to the Laplace-Belrtami eigen-basis, specified and fixed *a priori*. While this choice might be reasonable for near-isometric 3D shapes represented as triangle meshes, it does not allow to handle more diverse deformations classes of or significant noise in the data.

Inspired by the success and robustness of these techniques, we propose the first fully-differentiable functional maps pipeline, in which both the probe functions and the functional basis are learned from the data. Our key observation is that basis learning can be phrased as computing an embedding into a higher-dimensional space in which a non-rigid deformation becomes a *linear transformation*. This follows the functional map paradigm in which functional maps arising from pointwise correspondences must always be linear [39] and computing such a linear transformation is equivalent to solving the non-rigid correspondence problem. In the process, we also observe that training a network that aims to compute a *canonical* embedding, in which optimal correspondences are simple nearest neighbors, leads to a drop in performance. As we discuss below, this suggests that the additional degree of freedom, by learning a linearly-invariant embedding, helps to regularize the learning process and avoid overfitting in challenging cases. Finally, we demonstrate that our simple (but effective) formulation leads to accurate dense maps. The code, datasets and our pre-trained networks can be found online: https://github.com/riccardomarin/Diff-FMaps.

## 2 Related work

In addition to approaches mentioned above, here we briefly discuss previous works in the shape correspondence domain that are either closest to ours or most relevant for comparison and evaluation. We refer to the available surveys [5, 50] for a more complete overview.

**Functional maps** The core of our method is the functional maps framework originally proposed in [39] which formulates the correspondence problem in the functional domain instead of the classical matching between points. In the functional space, a correspondence can be represented by a small matrix encoded in a reduced basis and computed as the optimal transformation that aligns a given set of probe functions possibly with other regularization. This method inspired a large number of further extensions, including [38, 14, 47, 46] to name a few. A more general overview of this area can be found in [40]. In our paper we also exploit the link between the functional representation and the adjoint map that has been originally developed in [24].

The most common basis used in the functional map framework is given by the eigenfunctions of the Laplace-Beltrami operator, which can be seen as a natural extension of the Fourier basis to non-Euclidean domains [28, 53]. These basis functions are appropriate for shapes represented as triangle meshes, undergoing near-isometric deformations. Unfortunately, however, they can be highly unstable and difficult to estimate reliably for more general deformations and on point cloud data. Possible alternatives to this choice have been proposed in the literature such as [36, 37, 34]. These works try to recover the information lost by the low-pass representation of the truncated Fourier

basis but still suffer from the same underlying limitations. Several efforts have been made to apply the functional map framework to point cloud data, including [47, 35]. These works exploit existing discretizations of the Laplace-Beltrami operator on point clouds [4, 30], and present acceptable results under clean dense sampling but quickly deteriorate in more challenging scenarios.

Other works have been devoted to the selection of appropriate probe functions used to guide the computation of functional maps. Axiomatic descriptors such as HKS, WKS or SHOT [51, 3, 56] are widely used as probe functions together with supervised information such as segments and landmarks [17, 11]. More recently, an optimisation-based strategy has been proposed to compute optimal relative weights of probe functions [10], while a set of five automatically estimated stable landmarks has been used as probe functions for functional maps on human shapes in [32].

**Learning based methods for functional maps**    While early works in functional maps are purely axiomatic [26, 42, 37, 47], this framework has also recently been adapted to the learning setting. Specifically, starting with the seminal work of Deep Functional Maps [31], several methods have been proposed to *learn* optimal descriptors that can be used within the functional maps framework [22, 48, 13]. Most recently, it was demonstrated in [13] that the optimal descriptor (or probe) functions can be learned directly from the 3D coordinates of the shapes. This work has also shown that a functional map layer can help to regularize shape correspondence learning, leading to better results with less training data compared to state-of-the-art purely point-based methods [20]. Nevertheless, the approach of [13] is still tied to the choice of the Laplace-Beltrami eigenbasis and therefore lacks robustness in challenging non-isometric settings. Instead our fully learnable pipeline allows to benefit from the functional map regularization while being both robust and applicable to point cloud data.

**Other approaches**    A different line of work has also aimed to learn correspondences between 3D shapes by coordinate transfer [20, 21]. Other recent techniques also use geometric information through diverse convolution operations [61, 33, 18, 12, 59] and have demonstrated their effectiveness in 3D shape matching, typically by phrasing it as a dense segmentation problem. Learning for partial *rigid* alignment has been also proposed [60]. Correspondences can also be computed through finding a canonical embedding of the input. This idea has been developed for 2D images [8, 54] as well as 3D data [62]. The latter works, as many others in this domain, take advantage of point-based architectures such as PointNet [44] and its extensions [45, 2, 55] that provide a powerful way to learn signatures for point clouds, and that have been mainly exploited for shape classification but not yet for *smooth and consistent* dense non-rigid shape correspondence.

## 3    Background, motivation and notation

In this section, we give a brief overview of the functional map representation and correspondence pipeline. We then provide a general motivation behind our work and introduce the main notation that we adopt in the rest of the paper.

**Functional Maps**    We start by summarizing the functional maps framework. This formalism was initially developed for smooth surfaces, and most of the constructions have immediate analogues in the discrete setting when shapes are represented as triangle meshes. Note that we describe our learning-based pipeline adapted to point clouds in the following sections. Given a pair of shapes $\mathcal{X}$ and $\mathcal{Y}$, let $\mathcal{F}(\mathcal{X})$ and $\mathcal{F}(\mathcal{Y})$ denote the spaces of real-valued functions on $\mathcal{X}$ and $\mathcal{Y}$, respectively. A point-to-point map $T_{\mathcal{X}\mathcal{Y}} : \mathcal{X} \to \mathcal{Y}$ induces a functional correspondence $T^{\mathcal{F}}_{\mathcal{Y}\mathcal{X}} : \mathcal{F}(\mathcal{Y}) \to \mathcal{F}(\mathcal{X})$ via pull-back (notice that $T^{\mathcal{F}}_{\mathcal{Y}\mathcal{X}}$ goes in the opposite direction). If we approximate the space of functions in a given basis $\Phi_{\mathcal{X}}$ and $\Phi_{\mathcal{Y}}$ of size $k$, then $T^{\mathcal{F}}_{\mathcal{Y}\mathcal{X}}$ can be compactly represented by a matrix $C_{\mathcal{Y}\mathcal{X}}$ of size $k \times k$ that maps the coefficients of a function in the basis $\Phi_{\mathcal{Y}}$ to the coefficients of its image via $T^{\mathcal{F}}_{\mathcal{Y}\mathcal{X}}$ in the basis $\Phi_{\mathcal{X}}$. Specifically, if the basis is orthonormal, then the entries of this matrix have an explicit expression: $C_{\mathcal{Y}\mathcal{X}}(i,j) = < T^{\mathcal{F}}_{\mathcal{Y}\mathcal{X}}(\phi^{\mathcal{Y}}_j), \phi^{\mathcal{X}}_i >$, where $<,>$ denotes the functional inner product and $\phi^{\mathcal{X}}_i, \phi^{\mathcal{Y}}_j$ are the individual basis functions on $\mathcal{X}$ and $\mathcal{Y}$ respectively.

The most common choice for $\Phi_{\mathcal{X}}$ and $\Phi_{\mathcal{Y}}$ is the set of the eigenfunctions of the Laplace-Beltrami operator $\Delta$ associated with the $k$ eigenvalues with smallest absolute value. This choice was advocated in the original functional maps work [39] and then later used in virtually all follow-up approaches, including learning-based ones, e.g., [47, 38, 23, 31, 22] among others (see also [40] for an overview).

The Laplacian eigenfunctions naturally generalize the Fourier basis to non-Euclidean domains [53, 27, 28] and enjoy many similar properties such being ordered from low to higher frequencies, and spanning the space of $L^2$ functions. In practice, this basis can be computed efficiently on 3D triangle meshes via an eigen-decomposition of the standard cotangent Laplacian matrix [41].

A typical pipeline for solving the shape correspondence problem based on the functional map representation consists of the following steps [40, 46, 35, 47]: 1) Establish the basis functions, by computing the first $k$ Laplace-Beltrami eigenfunctions on $\mathcal{X}, \mathcal{Y}$ and store them as columns of matrices $\Phi_{\mathcal{X}}, \Phi_{\mathcal{Y}}$. 2) Compute probe functions $G_{\mathcal{X}}, G_{\mathcal{Y}}$ which are expected to be preserved by the underlying unknown map. 3) Compute the optimal functional map by solving:

$$C_{\mathcal{X}\mathcal{Y}} = \underset{C \in \mathbb{R}^{k \times k}}{\arg \min} \|C\Phi_{\mathcal{X}}^{\dagger}G_{\mathcal{X}} - \Phi_{\mathcal{Y}}^{\dagger}G_{\mathcal{Y}}\|_2 + E_{reg}(C). \quad (1)$$

Here $^{\dagger}$ denotes the Moore Penrose pseudo-inverse, so that, e.g., $\Phi_{\mathcal{X}}^{\dagger}G_{\mathcal{X}}$ represents the coefficients of the probe functions in the given basis. The second term in the sum is the regularization on the functional map which promotes some structural properties of the correspondence. For example a popular choice is $E_{reg}(C) = \|C\Delta_{\mathcal{X}} - \Delta_{\mathcal{Y}}C\|$, which enforces the commutativity between the functional map and Laplace-Beltrami operators (expressed in the respective basis), thereby promoting near-isometric point-to-point correspondences [39]. Finally, 4) refine the functional map computed in the previous step and convert it to a dense correspondence, e.g., via nearest neighbor search [39].

One of the advantages of this approach is that the optimization in step 3) can be performed efficiently since $C_{\mathcal{X}\mathcal{Y}}$ is a matrix of size $k \times k$, which is typically much smaller than the number of points.

**Limitations** The main limitations of this pipeline are two-fold: first, the quality of the map is strongly tied to the choice of probe functions, and second, the choice of the basis plays a fundamental role both for the expressive power and the accuracy of the final results. Several approaches have been proposed to learn the probe functions from data [31, 22, 48, 13]. However, as mentioned above, no existing methods have attempted to learn the basis. This is particularly problematic since, as we show below, as the Laplacian eigen-basis is not only tied to near-isometric deformations, even more fundamentally, it can only be reliably computed on shapes represented as triangle meshes. While some attempts (e.g., in [35, 47]) have been made to compute eigenfunctions using existing discretizations of Laplace-Beltrami operators on point clouds, e.g., [4, 30]. Nevertheless, in part due to the *differential nature* of the Laplacian, such discretizations cannot handle even mild levels of noise, in practice.

## 3.1 Motivation and Overview

Our main goal is to learn an optimal basis that can be used within the functional map pipeline on point cloud data. One possibility would be to use triangle meshes and learn a discretization of the Laplacian that would approximate the low frequency basis functions on point clouds. However, this requires differentiating through a sparse eigen-decomposition, which can be expensive and unstable.

Instead, we propose an end-to-end learnable pipeline that uses a dual point of view. We summarize our overall pipeline in Figure 1. Our first remark is that entries of the basis functions can be interpreted as an embedding of the original 3D shape into a higher $k$-dimensional space. Namely, each point $x \in \mathcal{X}$ gets associated with a $k$-dimensional vector $[\phi_1^{\mathcal{X}}(x), \phi_2^{\mathcal{X}}(x), \ldots, \phi_k^{\mathcal{X}}(x)]$. This is called the "spectral" embedding and it is well-known (see e.g., [49]) that when using the Laplacian basis on smooth surfaces, as $k \to \infty$ this embedding becomes injective, so that no two points can have the same associated vectors.

The spectral embedding plays a role in the conversion between functional and pointwise maps. The standard approach for this conversion [39] is by mapping Dirac $\delta_x$ functions associated with each point $x$ on the source shape and finding the nearest Dirac $\delta$ function on the target. Interestingly, $\delta_x$ is *not* a real-valued function but is rather a *distribution*, which acts on real-valued functions through inner products: $< \delta_x, f > = f(x)$. As functional maps are operators that map real-valued functions, in principle they *cannot* be used to transport Dirac $\delta$'s. To transport such distributions, a more sound approach is to use the *adjoint* operator of a functional map [24]. Surprisingly, although the notion of the adjoint has been studied , both its role and the limitations of functional maps in transferring $\delta$ functions seems to have been ignored in the functional maps literature so far. The adjoint operator is defined implicitly as follows: given a functional map $C_{\mathcal{Y}\mathcal{X}}$, its adjoint $A_{\mathcal{X}\mathcal{Y}}$ is defined so for any

pair of real-valued functions $f \in \mathcal{F}(\mathcal{X})$ and $g \in \mathcal{F}(\mathcal{Y}) : < C_{\mathcal{YX}}g, f > = < g, A_{\mathcal{XY}}f >$. We refer to the supplementary for a more complete treatment of the adjoint operator. Note that the adjoint operator: 1) associates functions in the opposite direction to that of the functional map, and 2) is defined using the $L_2$ inner products, and can thus be used to transport distributions. It is easy to see that the adjoint of the pull-back of a point-to-point map $T_{\mathcal{XY}}$ (see proof in the supplementary material) has the following nice property: $A_{\mathcal{XY}}\delta_x = \delta_{T_{\mathcal{XY}}(x)}$.

Finally, we note that the coefficients of Dirac $\delta$ function $\delta_x$ are precisely the vector of values $[\phi_1^{\mathcal{X}}(x), \phi_2^{\mathcal{X}}(x), \ldots, \phi_k^{\mathcal{X}}(x)]$. Moreover, the adjoint is a linear operator that associates $\delta$ functions with $\delta$ functions. As such, the adjoint can be seen as a linear transformation that aligns the spectral embeddings of $\mathcal{X}$ and $\mathcal{Y}$. We emphasize that the same *does not hold* for a functional map, in general.

This discussion implies that in the functional map framework, the basis can be interpreted as an embedding, and moreover the corresponding embeddings are related by a linear transformation, which is precisely the adjoint of the functional map.

**Strategy**   Our overall strategy is to mimic this construction using a learning-based approach. We propose to train a network that computes for each shape an embedding into some $k$ dimensional space, such that the embeddings of two shapes are related by a linear transformation. We then train a separate network that computes probe functions that can be used for establishing the optimal linear transformation at test time. Remarkably, this decomposition of the problem consistently outperforms a baseline approach that aims to compute a canonical embedding, in which correspondences can be obtained through nearest neighbor search directly. As described below, we attribute this primarily to the fact that learning a canonical embedding is a difficult problem, and splitting it into two parts (invariant embedding + transformation) helps to regularize the problem in challenging practical settings. Note that we use the term "basis" only by analogy with the Laplace-Beltrami eigenfunctions, and do not formally impose a basis structure on our learned set of functions.

## 4   Linearly-invariant embedding

In this section, we propose a novel learning strategy to generalize the functional maps framework to noisy and incomplete data.

We discretize a shape $\mathcal{X}$ as a collection of 3D points $x_i \in \mathbb{R}^3$ where $i \in \{1, \ldots, n_{\mathcal{X}}\}$. We collect these $n_{\mathcal{X}}$ points in a matrix $P_{\mathcal{X}} \in \mathbb{R}^{n_{\mathcal{X}} \times 3}$ such that the $i$-th row of $P_{\mathcal{X}}$ captures the 3D coordinates of $x_i$. We refer to the matrix $P_{\mathcal{X}}$ as the *natural* embedding of $\mathcal{X}$.

Given a pair of shapes $\mathcal{X}$ and $\mathcal{Y}$ our goal is to find a correspondence between them. This correspondence is a mapping between the points of $\mathcal{X}$ and the points of $\mathcal{Y}$. We denote a correspondence as a map $T_{\mathcal{XY}} : \mathcal{X} \to \mathcal{Y}$ such that $T_{\mathcal{XY}}(x_i) = y_j, \forall i \in \{1, \ldots, n_{\mathcal{X}}\}$ and some $j \in \{1, \ldots, n_{\mathcal{Y}}\}$. This map has a natural matrix representation $\Pi_{\mathcal{XY}} \in \mathbb{R}^{n_{\mathcal{X}} \times n_{\mathcal{Y}}}$ such that $\Pi_{\mathcal{XY}}(i, j) = 1$ if $T_{\mathcal{XY}}(x_i) = y_j$ and 0 otherwise.

Let $\Phi_{\mathcal{X}}$ and $\Phi_{\mathcal{Y}}$ denote the matrices, whose rows can be interpreted as embeddings of the points of $\mathcal{X}$ and $\mathcal{Y}$ as described in Section 3. Below we do not assume that $\Phi_{\mathcal{X}}$ and $\Phi_{\mathcal{Y}}$ represent the Laplacian eigenbasis, but consider general embeddings into some fixed $k$ dimensional space. Recall that in the formalism of functional maps, there must exist a linear transformation $A_{\mathcal{XY}}$ that aligns the corresponding embeddings. This can be written as: $A_{\mathcal{XY}}\Phi_{\mathcal{X}}^T = (\Pi_{\mathcal{XY}}\Phi_{\mathcal{Y}})^T$, where $\Pi_{\mathcal{XY}}$ is the binary matrix that encodes the correspondence between $\mathcal{X}$ and $\mathcal{Y}$. In the functional map framework, the linear transformation $A_{\mathcal{XY}}$ is precisely the adjoint operator, since $A_{\mathcal{XY}} = (\Phi_{\mathcal{X}}^+\Pi_{\mathcal{XY}}\Phi_{\mathcal{Y}})^T = C_{\mathcal{YX}}^T$ using the standard definition of a functional map $C_{\mathcal{YX}}$ [40].

Given $A_{\mathcal{XY}}$, we can estimate $\Pi_{\mathcal{XY}}$ by solving the following optimization problem:

$$\Pi_{\mathcal{XY}} = \arg\min_{\Pi} \|\Phi_{\mathcal{X}}A_{\mathcal{XY}}^T - \Pi\Phi_{\mathcal{Y}}\|_2. \tag{2}$$

Note that Eq. (2) can be solved in closed form by finding, for every row of $\Phi_{\mathcal{X}}A_{\mathcal{XY}}^T$, the closest row in $\Phi_{\mathcal{Y}}$ in the standard $L_2$ sense.

Based on Equation (2), our general goal is to train a network $\mathcal{N}$ that can produce for any shape $\mathcal{X}$ an embedding $\Phi_{\mathcal{X}}$ into a $k$-dimensional space, such that embeddings of every pair of shapes $\Phi_{\mathcal{X}}, \Phi_{\mathcal{Y}}$ are related by a linear transformation. In other words the network $\mathcal{N}$ must be able to transform a shape

from the original 3D space, in which complex non-rigid deformations could occur, to another space, in which transformations across shapes must always be linear. Interestingly, as we show below, the additional linear degree of freedom helps to regualize the learning procedure, achieving better results than simply learning a canonical embedding in which corresponding points are nearest neighbors.

## 4.1 Learning a linearly-invariant embedding

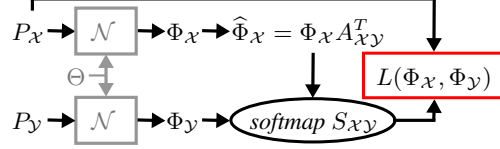

To learn a linearly-invariant embedding we first observe that for fixed matrices $\Phi_{\mathcal{X}}, \Phi_{\mathcal{Y}}$ the expression $\|\Phi_{\mathcal{X}} A_{\mathcal{X}\mathcal{Y}}^T - \Pi_{\mathcal{X}\mathcal{Y}}\Phi_{\mathcal{Y}}\|_2$ depends both on $A_{\mathcal{X}\mathcal{Y}}^T$ and $\Pi_{\mathcal{X}\mathcal{Y}}$, which can make training difficult. However, for a fixed correspondence matrix $\Pi_{\mathcal{X}\mathcal{Y}}$ the optimal matrix $A_{\mathcal{X}\mathcal{Y}}$ can be obtained in closed form simply as: $A_{\mathcal{X}\mathcal{Y}} = (\Phi_{\mathcal{X}}^+ \Pi_{\mathcal{X}\mathcal{Y}}\Phi_{\mathcal{Y}})^T$, which can be computed by solving a linear system of equations. Importantly, this procedure can be differentiated using the closed-form expression of derivatives of matrix inverses, which we exploit in our approach.

**Embedding network training** Given a set of training pairs of shapes $\mathcal{X}, \mathcal{Y}$ for which ground truth correspondences $\Pi_{\mathcal{X}\mathcal{Y}}^{gt}$ are known, our embedding network $\mathcal{N}$ computes an embedding $\Phi_{\mathcal{X}}, \Phi_{\mathcal{Y}}$ for each shape using a Siamese architecture with shared parameters. I.e., $\mathcal{N}_{\Theta}(P_{\mathcal{X}}) = \Phi_{\mathcal{X}}$ and $\mathcal{N}_{\Theta}(P_{\mathcal{Y}}) = \Phi_{\mathcal{Y}}$. We use the notation $\mathcal{N}_{\Theta}$ to highlight that this network has trainable parameters $\Theta$ which are shared across shapes. In the following we refer to this network as simply $\mathcal{N}$. The exact details of the architecture that we use are provided in the supplementary.

In order to define our loss we first compute the optimal linear transformation $A_{\mathcal{X}\mathcal{Y}} = (\Phi_{\mathcal{X}}^+ \Pi_{\mathcal{X}\mathcal{Y}}^{gt}\Phi_{\mathcal{Y}})^T$ and use it to obtain a *transformed* embedding $\widehat{\Phi}_{\mathcal{X}} = \Phi_{\mathcal{X}} A_{\mathcal{X}\mathcal{Y}}^T$. We then compare the rows of $\widehat{\Phi}_{\mathcal{X}}$ to those of $\Phi_{\mathcal{Y}}$ to obtain the *soft* permutation matrix $S_{\mathcal{X}\mathcal{Y}}$ that approximates the discrete mapping between the shapes in a differentiable way using the *softmax* operation (for completeness see details in the supplementary). Finally, we use the following loss to train the embedding network:

$$L(\Phi_{\mathcal{X}}, \Phi_{\mathcal{Y}}) = \frac{1}{n_\beta} \sum \|S_{\mathcal{X}\mathcal{Y}} P_{\mathcal{X}} - \Pi_{\mathcal{X}\mathcal{Y}}^{gt} P_{\mathcal{X}}\|_2^2. \tag{3}$$

Recall that $P_{\mathcal{X}}$ is the matrix encoding the 3D coordinates of the shape $\mathcal{X}$.

Intuitively, the main goal of the loss in Eq. (3) is to compare the ground truth correspondence $\Pi_{\mathcal{X}\mathcal{Y}}^{gt}$ to the computed softmap matrix $S_{\mathcal{X}\mathcal{Y}}$. An alternative would to use the geodesic distances as weights as done in [31], but the computation of the geodesic distances is expensive and unreliable in the context of point clouds. Other possible solutions are the direct Frobenius loss on the permutation matrix or a *multinomial regression loss* as done in e.g. [33, 43]. However, these losses do not involve the geometry and penalize incorrect correspondences independently of their proximity to correct ones. Instead, our loss penalizes incorrect correspondences based on the Euclidean distances of associated points. Moreover, Eq. (3) can be seen as the comparison between the action of the ground-truth functional map in the full basis and the action of the estimated functional map on a specific set of functions that completely describe the geometry of the data. As such, our loss is efficient, takes the geometry into account, and is directly related to the functional map formalism.

## 4.2 Learning the optimal transformation

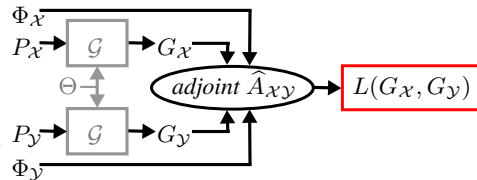

As mentioned above, we train our approach in two stages: first we train an embedding network using the loss described in Section 4.1. We then train a separate network that aims to compute an optimal linear transformation between the embeddings, which can be used to compute correspondences at test time. Our observation is that this linear transformation can be obtained given enough constraints, by solving a linear system. Therefore, following the ideas in Deep Functional Maps [31] our second network $\mathcal{G}$ takes as input the natural embedding of a shape and outputs a set of $p$ "probe" functions via $\mathcal{G}_{\Theta}(P_{\mathcal{X}}) = G_{\mathcal{X}}$ and $\mathcal{G}_{\Theta}(P_{\mathcal{Y}}) = G_{\mathcal{Y}}$ using shared trainable parameters $\Theta$. We then minimize the following loss:

$$L(G_{\mathcal{X}}, G_{\mathcal{Y}}) = \|A_{\mathcal{X}\mathcal{Y}}^{gt} - \widehat{A}_{\mathcal{X}\mathcal{Y}}\|_2. \tag{4}$$

Here $A_{\mathcal{XY}}^{gt}$ is the ground truth linear transformation between the learned embeddings $A_{\mathcal{XY}}^{gt} = (\Phi_{\mathcal{X}}^{+} \Pi_{\mathcal{XY}}^{gt} \Phi_{\mathcal{Y}})^T$ whereas $\widehat{A}_{\mathcal{XY}} = \left( (\Phi_{\mathcal{Y}}^{\dagger} G_{\mathcal{Y}})^T \right)^{\dagger} (\Phi_{\mathcal{X}}^{\dagger} G_{\mathcal{X}})^T$. This equation arises from the fact that if $A_{\mathcal{XY}}$ is the adjoint that aligns the embeddings then $A_{\mathcal{XY}}^T$ is a functional map from $\mathcal{Y}$ to $\mathcal{X}$ which implies that $A_{\mathcal{XY}}^T \Phi_{\mathcal{Y}}^{\dagger} G_{\mathcal{Y}} = \Phi_{\mathcal{X}}^{\dagger} G_{\mathcal{X}}$ whenever $G_{\mathcal{X}}, G_{\mathcal{Y}}$ are corresponding functions. Please see the supplementary material for a more detailed discussion.

### 4.3 Test phase

Once we train these two networks, we can estimate the correspondence between an arbitrary pair of point clouds $\mathcal{X}$ and $\mathcal{Y}$ in four steps: (1) compute the embeddings $\Phi_{\mathcal{X}}$ and $\Phi_{\mathcal{Y}}$ using the embedding network $\mathcal{N}$; (2) compute the set of probe functions, $G_{\mathcal{X}}$ and $G_{\mathcal{Y}}$ using the network $\mathcal{G}$; (3) solve for the linear transformation $A_{\mathcal{XY}}$ using the expression given for $\widehat{A}_{\mathcal{XY}}$ above; (4) estimate for the correspondence $\Pi_{\mathcal{XY}}$ via nearest neighbor search as described in Eq. (2).

**Discussion** While the basis and probe function networks appear similar as they both output a matrix, they are different in their losses and, as consequence, in the task that they solve. Our first linearly-invariant embedding (basis) network aims to output a representation in $k$ dimensions so that different shapes share the same structure up to rotation and non-uniform scaling. Further, our loss in Eq (3) promotes continuity of the embedding with respect to the original shape coordinates. In contrast, the descriptor network aims to find a small set of reliable descriptors that can establish the linear transformation in the $k$ dimensional space. Our strategy is different from a network which would aim to find an embedding where correspondences are directly obtained as nearest neighbors (we call this option a "universal embedding"), as such a network would have to disambiguate each point directly. Instead, by first obtaining a smooth embedding and then using a small number of salient feature descriptors (probe functions in our case) our approach allows to find a dense correspondence even in challenging cases, in which individual points may not be easy to distinguish.

## 5 Experiments

We evaluate our pipeline on the correspondence problem between non-rigid 3D point clouds in the challenging class of human models. We use this class because of the availability of data and baselines for comparison but stress that our method is general and can be applied to any shape category.

**Architecture and parameters** Both of our networks $\mathcal{N}$ and $\mathcal{G}$ are built upon the PointNet architecture [44]. For our experiments we train over 10K shapes from the SURREAL dataset [57], resampled at 1K vertices. We learn a $k = 20$ dimensional embedding (basis) and $p = 40$ probe functions for each point cloud. We report in Supplementary Materials the complete description of the architectures and the training data.

### 5.1 Non-isometric pointclouds

We consider a first test set composed by the 100 shapes from the FAUST dataset [6] (10 different subjects in 10 different poses). We treat each shape as an unorganized point cloud selecting only $1K$ of its vertices and discarding mesh connectivity. We generate a second test set perturbing the first one with Gaussian noise. In both test sets, we deal with non-isometric pairs (different subjects) and strong non rigid deformations (different poses). The second one is particularly challenging because it ruins the underlying shape structure. As competitive baselines we consider *universal embeddings* (Uni20 and Uni60) obtained with the same architecture we used for $\mathcal{N}$ by learning 20 and 60 basis respectively, but enforcing the optimal linear transformation to be identity. We also compare our method with the standard functional maps, with 5 ground-truth landmarks (FMAP), the recent state-of-the-art methods (GFM) [13], and finally against 3D-CODED [20] (3DC). For the GFM and FMAP methods we also compare to a version refined with ZoomOut [35] (FMAP+ZOO, GFM+ZOO). For the methods that require the LBO basis, we adopt the estimation of LBO for point clouds proposed in [9]. As can be seen in Figure 2, we outperform the baseline and all the competitors including the the state-of-the-art methods GFM and 3DC in both the considered scenarios. We stress that both [20] and [13] are very recent highly complex state-of-the-art methods, with e.g. [20] being directly adapted to point clouds with an expensive test-time post-processing. Our method achieves

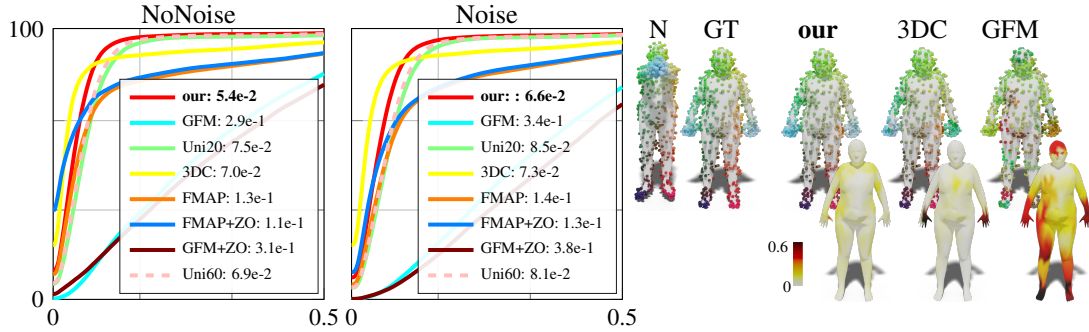

Figure 2: The evaluation of the correspondence for point clouds generated from the FAUST dataset without or with additional noise. On the left, cumulative curves with mean error in the legends. On the right, a qualitative example in Noise setup, with the related hotmap error.

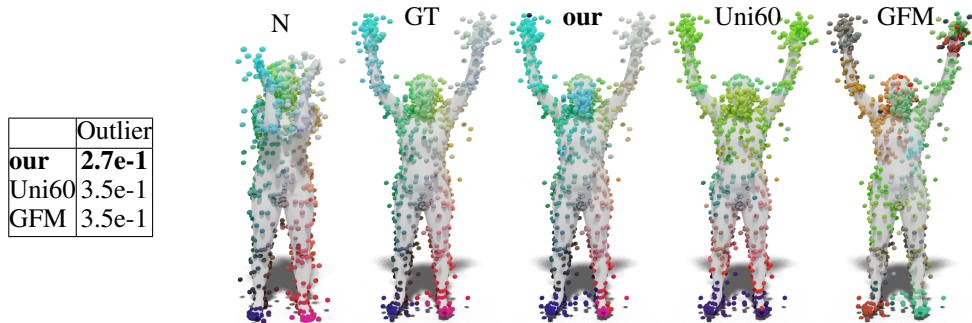

|       | Outlier |
|-------|---------|
| **our** | **2.7e-1** |
| Uni60 | 3.5e-1 |
| GFM   | 3.5e-1 |

Figure 3: Qantitative results on 100 pairs of the test set with 30% outlier points, compared to the baselines, with a qualitative example.

state-of-the-art results without any additional post-processing. Further robustness of our method is illustrated in Figure 3, where we evaluate our networks trained on clean data, on the FAUST test set augmented with outliers points. Our method shows significant resilience and outperforms competing methods in this challenging setting, despite not being presented with outlier data at training time.

In addition, in Figure 4 we also visualize a correspondence, computed using our network, between a pair of real-world scans taken from the *Scan the world* project collection [1]. The presence of significant topological changes, partiality, clutter, non-isometry and self-intersections represent a significant challenge. Despite this, our method, shows remarkable resilience and provides a reliable result even without retraining or post-processing. We provide more illustrations on real scans in the Supplementary Materials.

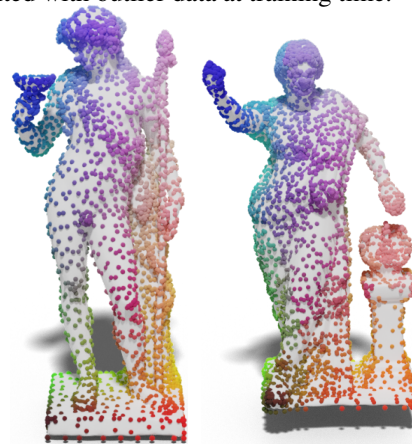

Figure 4: A qualitative example of matching between two statues. Despite the presence of clutter, partiality and non-isometry our point cloud-based approach shows resilience.

## 5.2 Fragmented partiality

Finally, we compare our approach, the universal embedding and the LBO basis (LBO) in an extreme scenario. We compute a correspondence between each of the 100 full shapes from FAUST and a fragmented version that consists of several small disconnected components. This experiment tests how each basis is affected by heavy loss of geometry. Fixing a basis, we evaluate 1) the matching using a ground-truth transformation to retrieve the optimal linear transformation, on the left of Figure 5; 2) the correspondence estimated with the best pipeline for the given basis, on the right. The average geodesic errors are reported in the legends. In 2) for LBO we consider partial functional maps (PFM)

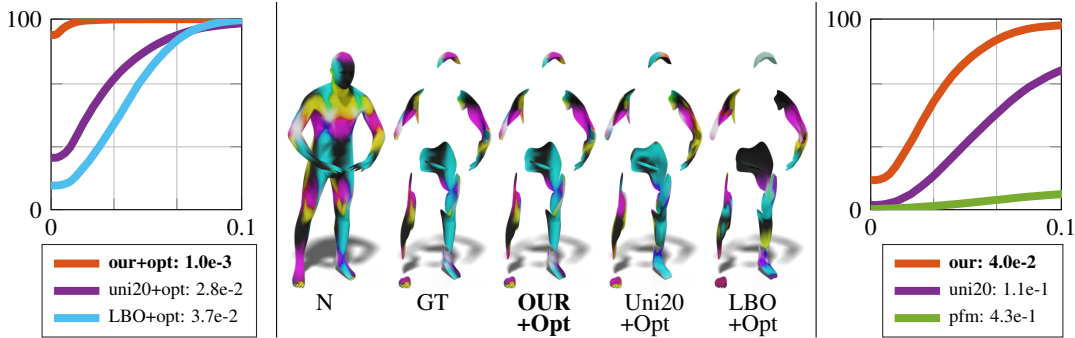

Figure 5: Partial setup. The shape is matched with a fragmented version of itself. We show the amount of information lost by the basis due to surface destruction and compare our method to baseline and partial functional map (pfm) [47]. More details in the main text.

[47], which extends the functional maps framework to partial cases. In the middle we visualize a qualitative comparison on one of the 100 pairs tested, where the correspondence in encoded by the color transfer. We highlight that it is not always possible to have a transformation that produces a perfect matching. LBO+opt and PFM suffer from the significant sensitivity of the LBO to partiality and topological noise. The universal embedding shows also a significant loss of information. With the linear invariant embedding it is still possible to retrieve good information and to generalize to corrupted data that are completely unseen during training.

# 6 Conclusion

In this paper, we presented an extension to the functional maps framework by replacing the standard Laplace-Beltrami eigenfunctions with learned functions. We achieve this by learning an optimal linearly-invariant embedding and a separate network that aligns embeddings of different shapes.

While general, our approach still assumes that the input data poses a "natural" embedding in 3D making it yet not applicable to data such as graphs. Moreover, we do not exploit the mesh structure that *might* be available in certain cases. Combining our method with a mesh-aware approach is an interesting direction for future work. Our preliminary investigation outperforms the competitors in challenging scenarios. We believe that these results only scratch the surface and can pave the way to future work on invariant embeddings for shape correspondence and other related problems.

# 7 Broader Impact

Computing reliable correspondences is a problem that arises in many scientific disciplines and practical scenarios including medical imaging and industrial quality control (for detecting anomalies and performing repair and analysis), as well as 3D animation and texture transfer, statistical shape analysis, and even personalized medicine (with accurate detection of measurements often performed by template matching). Our novel fully trainable pipeline paves the way to more accurate results with direct practical applications in all of these fields, especially as it is applicable to arbitrary 3D shapes and deformations, unlike many existing methods which are specific e.g. to humans or near-isometries. This has the potential to replace highly specific axiomatic methods and tedious manual intervention. Finally, our insights can shed light on the structure of functional maps and shape analysis more broadly. We do not see any ethical issue with the proposed method, at least no ethical issues may be caused by our method as it is.

# 8 Acknowledgements

The authors would like to thank the anonymous reviewers for their detailed feedback and suggestions. Parts of this work were supported by the KAUST OSR Award No. CRG-2017-3426, the ERC Starting Grant No. 758800 (EXPROTEA) and the ANR AI Chair AIGRETTE.

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
