[Supplementary Material]

# Supplementary Materials: Correspondence learning via linearly-invariant embedding

**Riccardo Marin**\*
University of Verona
riccardo.marin_01@univr.it

**Marie-Julie Rakotosaona**\*
LIX, Ecole Polytechnique, IP Paris
mrakotos@lix.polytechnique.fr

**Simone Melzi**
LIX, Ecole Polytechnique, IP Paris
Sapienza University of Rome
melzi@di.uniroma1.it

**Maks Ovsjanikov**
LIX, Ecole Polytechnique, IP Paris
maks@lix.polytechnique.fr

In this document, we collect all the discussions, proofs and results that due to the limited number of pages available did not found space in the main manuscript.

## 1 Adjoint operator definition and properties

In this section, we provide a concise description of the adjoint operator and its relation to the transfer of Dirac delta functions and functional maps. Note that the adjoint operator of functional maps has been considered, e.g., in [1] although its role in delta function transfer was not explicitly addressed in that work.

### 1.1 Formal definition of the Adjoint operator

Suppose we have a pointwise map $T_{\mathcal{X}\mathcal{Y}} : \mathcal{X} \to \mathcal{Y}$ between two smooth surfaces $\mathcal{X}, \mathcal{Y}$. Then we will denote $T_{\mathcal{Y}\mathcal{X}}^{\mathcal{F}}$ the functional correspondence defined by the pull-back: $T_{\mathcal{Y}\mathcal{X}}^{\mathcal{F}} : f \to f \circ T_{\mathcal{X}\mathcal{Y}}$, where $f : \mathcal{Y} \to \mathbb{R}$ and $f \circ T_{\mathcal{X}\mathcal{Y}} : \mathcal{X} \to \mathbb{R}$ such that $f \circ T_{\mathcal{X}\mathcal{Y}}(x) = f(T_{\mathcal{X}\mathcal{Y}}(x))$ for any $x \in \mathcal{X}$.

The *adjoint functional map operator* $A_{\mathcal{X}\mathcal{Y}}$ is defined implicitly through the following equation:

$$< A_{\mathcal{X}\mathcal{Y}}g, f >_{\mathcal{Y}} = < g, T_{\mathcal{Y}\mathcal{X}}^{\mathcal{F}}f >_{\mathcal{X}} \quad \forall \ f : \mathcal{Y} \to \mathbb{R}, \ g : \mathcal{X} \to \mathbb{R}. \quad (1)$$

Here we denote with $<,>_{\mathcal{X}}$ and $<,>_{\mathcal{Y}}$ the $L^2$ inner product for functions respectively on shape $\mathcal{X}$ and $\mathcal{Y}$. The adjoint always exists and is unique by the Riesz representation theorem (see also Theorem 3.1 in [1]).

### 1.2 Adjoint operator and delta functions

As mentioned in the main manuscript, the adjoint can be used to map *distributions* (or generalized functions), which is particularly important for mapping points represented as Dirac delta functions.

Recall that $\forall y \in \mathcal{Y}$, a Dirac delta function $\delta_y$ is a distribution such that, by definition, for any function $f$ we have $< \delta_y, f >_{\mathcal{Y}} = f(y)$.

**Theorem 1.** *If $A_{\mathcal{X}\mathcal{Y}}$ is the adjoint operator associated with a point-to-point mapping $T_{\mathcal{X}\mathcal{Y}}$ as in Eq. (1), then $A_{\mathcal{X}\mathcal{Y}}\delta_x = \delta_{T_{\mathcal{X}\mathcal{Y}}(x)}$.*

*Proof.* Using Eq. (1) we get:

$$< A_{\mathcal{XY}}\delta_x, f >_{\mathcal{Y}} = < \delta_x, T^{\mathcal{F}}_{\mathcal{YX}} f >_{\mathcal{X}} = < \delta_x, f \circ T_{\mathcal{XY}} >_{\mathcal{X}} \qquad (2)$$
$$= f(T_{\mathcal{XY}}(x)). \qquad (3)$$

Therefore, $A_{\mathcal{XY}}\delta_x$ equals some distribution $d$ such that $< d, f >_{\mathcal{Y}} = f(T_{\mathcal{XY}}(x))$ for any function $f : \mathcal{Y} \to \mathbb{R}$. By uniqueness of distributions this means that: $A_{\mathcal{XY}}\delta_x = \delta_{T_{\mathcal{XY}}(x)}$. $\qquad \square$

In other words, the previous derivation proves that, unlike a functional map, **the functional map adjoint always maps delta functions to delta functions**.

## 1.3 Relation between the functional maps and the adjoint operator in the discrete setting

Here we assume that the two shapes are represented in the discrete setting, with two embeddings $\Phi_{\mathcal{X}}, \Phi_{\mathcal{Y}}$, and a pointwise map $\Pi_{\mathcal{XY}}$, using the notation from the main paper. Our goal is to establish the relationship between the functional map matrix and the linear operator, which aligns the two embeddings.

Given two embeddings $\Phi_{\mathcal{X}}, \Phi_{\mathcal{Y}}$ and a pointwise map $\Pi_{\mathcal{XY}}$ we would like to find a linear transformation $A_{\mathcal{XY}}$ such that:

$$A_{\mathcal{XY}}\Phi_{\mathcal{X}}^T = (\Pi_{\mathcal{XY}}\Phi_{\mathcal{Y}})^T, \text{ or equivalently} \qquad (4)$$
$$\Phi_{\mathcal{X}} A_{\mathcal{XY}}^T = \Pi_{\mathcal{XY}}\Phi_{\mathcal{Y}} \qquad (5)$$

Formulating this as a least squares problem we get:

$$\min_{A} \|\Phi_{\mathcal{X}} A_{\mathcal{XY}}^T - \Pi_{\mathcal{XY}}\Phi_{\mathcal{Y}}\|_2, \qquad (6)$$

from which the solution is given by:

$$A = \left(\Phi_{\mathcal{X}}^{\dagger}\Pi_{\mathcal{XY}}\Phi_{\mathcal{Y}}\right)^T \qquad (7)$$

Recall that a functional map induced by $\Pi_{\mathcal{XY}}$ is defined as $C_{\mathcal{YX}} = \Phi_{\mathcal{X}}^{\dagger}\Pi_{\mathcal{XY}}\Phi_{\mathcal{Y}}$. Therefore, we can write: $A_{\mathcal{XY}} = C_{\mathcal{YX}}^T$. In other words, in the discrete setting the adjoint is nothing but the transpose of the functional map in the opposite direction.

## 1.4 Probe function constraints

Below we derive the relation between the probe function constraints for functional maps and those for the adjoint operator used in our approach, as described in Section 4.2 of the main paper. Here we derive the formula used in the main manuscript directly below Eq. (4).

In the main paper (Eq. (1) of the main manuscript) we wrote the following basic optimization problem for estimating functional maps:

$$C_{\mathcal{XY}} = \underset{C \in \mathbb{R}^{k \times k}}{\arg\min} \|C\Phi_{\mathcal{X}}^{\dagger}G_{\mathcal{X}} - \Phi_{\mathcal{Y}}^{\dagger}G_{\mathcal{Y}}\|_2 + E_{reg}(C). \qquad (8)$$

Inverting the role of $\mathcal{X}$ and $\mathcal{Y}$ and removing the regularization we obtain:

$$C_{\mathcal{YX}} = \underset{C \in \mathbb{R}^{k \times k}}{\arg\min} \|C\Phi_{\mathcal{Y}}^{\dagger}G_{\mathcal{Y}} - \Phi_{\mathcal{X}}^{\dagger}G_{\mathcal{X}}\|_2. \qquad (9)$$

This implies that the optimal $C_{\mathcal{YX}}$ can be found as the solution of $C_{\mathcal{YX}}\Phi_{\mathcal{Y}}^{\dagger}G_{\mathcal{Y}} = \Phi_{\mathcal{X}}^{\dagger}G_{\mathcal{X}}$. This is equivalent to $(\Phi_{\mathcal{Y}}^{\dagger}G_{\mathcal{Y}})^T C_{\mathcal{YX}}^T = (\Phi_{\mathcal{X}}^{\dagger}G_{\mathcal{X}})^T$ that can be solved as a least squares problem:

$$C_{\mathcal{YX}}^T = \left((\Phi_{\mathcal{Y}}^{\dagger}G_{\mathcal{Y}})^T\right)^{\dagger} (\Phi_{\mathcal{X}}^{\dagger}G_{\mathcal{X}})^T. \qquad (10)$$

From the equation $A_{\mathcal{XY}} = C_{\mathcal{YX}}^T$ we can conclude that:

$$A_{\mathcal{XY}} = \left((\Phi_{\mathcal{Y}}^{\dagger}G_{\mathcal{Y}})^T\right)^{\dagger} (\Phi_{\mathcal{X}}^{\dagger}G_{\mathcal{X}})^T. \qquad (11)$$

This is precisely the equation used in the main manuscript directly below Eq. (4).

This provides an explicit connection between the functional map and the linear transformation that we are optimizing for.

To summarize, one advantage of the adjoint is that it can be used to map *distributions* and not just functions. In particular, unlike a functional map, the functional map adjoint always maps delta functions to delta functions. At the same time, similarly to functional maps, it also allows estimation via probe functions and a solution of a linear system. For this reason, despite the strong relation with functional maps, the adjoint is better suited for estimating the correspondence.

## 2 Implementation details

### 2.1 Training set details

**Pre-processing** In our analysis, we consider shapes that are centered in the origin of $\mathbb{R}^3$ and scaled with uniform unit area. These requirements are not strong and every input shape can be easily pre-processed to satisfy these properties.

### 2.2 The softmax operation

We compute the *soft* permutation matrix as follows:

$$(S_{\mathcal{Y}\mathcal{X}})_{ij} = \frac{e^{-\|\widehat{\Phi}_{\mathcal{X}}^i - \Phi_{\mathcal{Y}}^j\|_2}}{\sum_{k=1}^{n_{\mathcal{Y}}} e^{-\|\widehat{\Phi}_{\mathcal{X}}^i - \Phi_{\mathcal{Y}}^k\|_2}} \tag{12}$$

where $n_{\mathcal{Y}}$ is the number of points of $\mathcal{Y}$ and $\Phi_{\mathcal{X}}$ and $\Phi_{\mathcal{Y}}$ are learned embeddings.

### 2.3 Architecture description

We describe our complete pipeline in figure 1. The invariant embedding network and probe function network are built with the semantic segmentation architecture of PointNet [2].

### 2.4 Relation to the universal embedding network

In figure 2, we show the training curves of the universal embedding model and the linearly-invariant embedding model. We observe that learning the linearly-invariant embedding leads to faster learning and a lower loss. It confirms that the linearly-invariant embedding simplifies and is more adapted to the correspondence task.

## 3 Additional results and visualizations

We show some other results on noisy point clouds in Figure 3. We provide an example of our networks output over a couple of FAUST shapes at high-resolution ( 160K vertices). We show the 20 basis in Figures 4 and 5, and the 40 descriptors in Figures 6, 7 and 8. Finally, we show further statues examples in Figure 9. We would remark that our method consider only the points coordinates; we color the surface for better visualization.

$\mathcal{X}$

nx3

invariant embedding network

probe functions network

$\Phi_{\mathcal{X}}$

nx20

nx40

$G_{\mathcal{X}}$

x

$\hat{A}_{\mathcal{X}\mathcal{Y}}$

$\Phi_{\mathcal{X}}\hat{A}_{\mathcal{X}\mathcal{Y}}^T$

nx20

20x20

$\Pi_{\mathcal{X}\mathcal{Y}}$

nxn

$\mathcal{Y}$

nx3

invariant embedding network

probe functions network

$\Phi_{\mathcal{Y}}$

nx20

nx40

$G_{\mathcal{Y}}$

nx20

$\Phi_{\mathcal{Y}}$

Figure 1: Method pipeline

2.89e+6
2.87e+6
2.85e+6
2.83e+6
2.81e+6

200k    600k    1M    1.4M

Uni test
Uni train
Invariant emb. test
Invariant emb. train

Figure 2: Comparison of the linearly-invariant embedding model and the universal embedding model training curves.

## Footnotes

\*denotes equal contribution.

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

Figure 3: Comparisons on noisy point clouds. 3DC major artifacts are over the hands and in some cases it confuses left and right. GFM suffers from the quality of the point clouds basis estimation.

$\Phi_X$  $\Phi_X A_{XY}$  $\Phi_Y$  $\Phi_X$  $\Phi_X A_{XY}$  $\Phi_Y$

Figure 4: Basis from 1 to 10

$\Phi_X$  $\Phi_X A_{XY}$  $\Phi_Y$  $\Phi_X$  $\Phi_X A_{XY}$  $\Phi_Y$

Figure 5: Basis from 11 to 20

Figure 6: Descriptors from 1 to 18

Figure 7: Descriptors from 19 to 36.

Figure 8: Descriptors from 37 to 40.

Figure 9: More qualitative results between statues couples.