[Reviews · NeurIPS 2020]

Review 1

Summary and Contributions: This paper aims to find one-to-one correspondences for two non-rigid 3D points clouds. The main contribution would be learning a transformation matrix to transform point-net learned point-wise features from one set to another. Experiments show the effectiveness of the proposed method

Strengths: 1) Sound theoretical explanation of the proposed method [Figure1 in the Supplementary Material] 2) Simply but effective modification to existing functional map representation and correspondence pipeline.

Weaknesses: My main concerns are on the experiments as I can get a clear picture of the pipeline in Figure1 in the Supplementary Material: 1) is the effectiveness of the additional learned transformation matrix (A_{XY}) sufficiently validated? Though authors include a universal embedding baseline (Line 295), and report its performance in Figure 2, I would say the performance gap between the proposed method and the universal embedding baseline is small. Furthermore, the universal embedding baseline only has one branch point-net to learn a 20-dim feature, instead, the proposed method has two branches (one branch to learn a 20-dim feature and the other to learn a 40-dim probe function). A correct way to include universal embedding baseline is to increase the feature dimension to 60 (20+40). As such, the ablation study concerning the number of feature and probe function dimensions should also be done. I would guess if the dimension is set to for example 128, the additional linear transformation matrix (A_{XY}) may not work. 2) Add outliers in experiments. For real-world scenario, there are outliers for both sets [source and target]. Therefore, the ablation study concerning the number/ratio of outlier 3D points should be done. [Both figure 2 and figure 3 do not have outlier 3D points] Minor points: 1) Line 277-278: [Instead, by first obtaining a smooth embedding and then using a small number of salient feature descriptors (probe functions in our case)] I don't think the effort to learn salient feature descriptors can be ignored. Refer to Figure 1 in the Supplementary Material, a branch to learn a point-wise probe function is added, with 40-dim features (doubles the dimension of embeddings from the invariant embedding network ). 2) Add x/y labels to figure 2 and 3.

Correctness: Yes

Clarity: Yes

Relation to Prior Work: Yes

Reproducibility: Yes

Additional Feedback: Authors did a good job to reply to my first-round comments and promise to update the paper, I do not have further questions.


Review 2

Summary and Contributions: The paper tackles an important problem in 3D shape matching: finding dense correspondences between 3D point clouds of non-rigid shapes. Following up on recent learning based approaches, that focused on learning shape descriptors and integrate the functional maps framework into a deep network, here the authors propose to also learn the basis functions instead of computing them manually. This has the advantage of promoting these bases to adhere to the linear relation between a pair of matching shapes. At the same time, it mitigates the errors which may come from pointcloud computation of laplacian eigen functions.

Strengths: + The key insight in this paper is nicely summarized by the authors in the sentence: "basis learning can be phrased as computing an embedding into a higher-dimensional space in which a non-rigid deformation becomes linear transformation". In my view, this is a nice distillation of the governing phenomenon behind the success of functional maps. This phrasing gives a natural criterion by which to train a data-driven pipeline. + The paper is very nicely written. It is easy to follow and has the right level of concision. + The two-branch design of the network, where one learns a mapping from 3D to spectral embedding and the other commutes a "probe functions" makes sense to me, and supported by meaningful loss functions.

Weaknesses: Weaknesses: - Equation (3) is measuring euclidean distance. While the authors nicely summarize the pros and cons of this design choice, an ablation on other choices would be beneficial. - linearly Invariant embedding network: First, i find the terminology is confusing. I believe the authors mean that the embedding of a pair of shapes is the same, up to a linear transformation but the term can be understood as an embedding that is invariant to linear transformations. My second concern is regarding the design of the network. The main paper points to the supplementary where it is described as: "semantic segmentation architecture of PointNet". There's really no need to refer the reader to the supp for a single sentence. Also, a pointNet is sensitive to transformations, both rigid and non-rigid. This calls for at least a discussion on how does this influence the proposed solution. - the method pipeline in the supp is describing the inference; however for training different Axy are used. The groundtruth Axy is used for the loss in equation (3). Minor: - A_xy is defined twice in the same way (once before eq (3) and once after eq(4) ) but once is called "ground-truth" and the second time it is not. - the paper does not discuss limitations. I'm curiious to know why, for example, such a severe downsampling to 1K points is needed? also, why were only 20 eigen functions used? was this same configuration used in the competing methods (e.g. was functional maps solved with only 20 evecs??). I suspect that the many pseudo-inverse operations may introduce a computational burden -- is this the reason for using so few evecs? - The usage of only 1K points also makes the comparison with sota methods problematic. For example, results on the official FAUST benchmark are not reported which is uncommon for works in the field. - The method in [30] is mentioned as the basis for this work however no comparison with it is shown. - I'm curious to know more about the properties of the learned "bases". For example, if only the first loss is used with non-leanred descriptors -- what is the performance? - partiality is mentioned a few times. I'd like to see how does partiality influence the learned basis? is it somehow less sensitive to boundary effects? what if different subsets of 1K are taken from the shape -- can the authors show that the basis is not sensitive to sampling density for example? - many descriptors are in fact functions of laplacian eigen functions like HKS and WKS. Did you try using the learned basis as input for the learned descriptors? or possibly together with the 3D pointcloud? typos: line 32: Laplace-Belrtami citations 26, 27 are repeated references: - for partial matching the PFM is mentioned. Here are a few follow up works in this domain which the authors may find useful * Fully Spectral Partial Shape Matching * Non-Rigid Puzzles * The Whole Is Greater Than the Sum of Its Nonrigid Parts - As this work can be views as learning an operator from shape, this work does the inverse: OperatorNet: Recovering 3D Shapes From Difference Operators

Correctness: See comments above

Clarity: Yes, See comments above

Relation to Prior Work: See comments above

Reproducibility: Yes

Additional Feedback: After reading the authors response, i still have several concerns: 1) regarding the first comment in the rebuttal i would like to point out that the functional maps framework has been used for pointclouds by computing the knn graph laplacian. See for example the experiments in "Partial Functional Correspondence" by Rodola et al. 2) While I agree that given the existence of multiple unsupervised approaches, these were introduced for meshes and not point clouds. Thus i actually don't think that "The main limitations of our method are its global and supervised nature. " -- but rather more concerned about the other questions which remain unanswered (number of evecs, and applicability to real scans, etc). I think these could really strengthen the papers. In particular results on FAUST benchmark and partiality results on the "cuts and holes" benchmark would be very helpful to appreciate the applicability of the method. 3) It is very hard to interpret the descriptors from the visualization in the supp. Perhaps some visualization of their uniqueness and localization would be helpful (choose a point on the shape and color the shape according to the similarity of all other points to it). Also, since the network is trained in two stages i wonder -- what is the result if using the learned "bases" with non-learned e.g. SHOT descriptors? it was also shown that naive pointnet may struggle in beating SHOT and special care has to be taken to learn good descriptors from scratch (see Continuous Geodesic Convolutions for Learning on 3D Shapes for example). To summarize, I think the direction proposed in this work is good and the results are promising. I believe that if the authors will consider the suggestion given here and improve the manuscript this can become a very strong paper however at its current state it is still not ready for publication.


Review 3

Summary and Contributions: The authors propose a fully differentiable pipeline for estimating correspondence between 3D point clouds based on the functional maps framework. Rather than using Laplace-Beltrami eigenfunctions their method learns the basis from data aiming to improve robustness and accuracy for challenging correspondences and resulting in an end-to-end trainable functional map-based correspondence approach. We authors argue that their approach achieves state-of-the-art results on challenging non-rigid 3D point cloud correspondence applications.

Strengths: Pro * Sound and very systematically derived approach * Very well formulated and structured text * Well structured and informative supplementary * Interesting and relevant research problem and significant results * Appropriate discussion of related work

Weaknesses: Con * The conducted experiment as well as its discussion appear a bit early and superficial. I would have liked more extensive experimentation, in-depth analysis of the proposed approach, and discussion of the results. * Minor: Figure 2+3 do not properly show the measured metric (mean error)

Correctness: * applied method is sound and very systematically derived * I would have liked to see more details regarding evaluation (the supp. provides few additional information in this regard)

Clarity: * the text is very will structured and clearly written

Relation to Prior Work: * the relation the previous work is well discussed and the text substantially refers to relevant related work throughout

Reproducibility: Yes

Additional Feedback: Rebuttal: the author did a good job in addressing comments and I still support this paper.


Review 4

Summary and Contributions: Authors proposed to register two point clouds (estimating correspondences) via a learning based functional map approach. The authors employ two networks, one to learn an embedding and the other to learn the probe functions. The approach is evaluated on trained on human pose point cloud dataset SURREAL and tested on FAUST dataset.

Strengths: 1. The overall paper is well written. 2. The overall framework is relatively straight forward. I do agree with the general approach of splitting the task into learning the embedding and then the transformation. 3. Method shows good results for aligning point clouds of different human poses.

Weaknesses: 1. The authors propose to train two different networks to learn the embedding and transformation separately with two PointNets (as stated in Supp Mat) whereas previous works directly learn the transformation to align the two point clouds. Can the authors elaborate on model and time complexity compared to previous works? 2. If the transformation A_{x, y} can be obtained directly using closed form from the correspondence matrix \Pi_{x, y}, why do we need to learn it in Sec. 4.2? 3. Based on the description of the framework, do the embedding network need to be trained first? Then it is used to train the network to learn the transformation? If so, is it possible that biases or even errors learned in the first will propagate to the second network? 4. The approach is only evaluated on one dataset for human poses. How well does this generalize to other unseen shapes? Will biases of human (generally standing upright) shapes cause problems if one is given point clouds of more irregular shapes like dogs, and cats? 5. Missing hyper parameters (learning rate, optimization method) to replicate results

Correctness: Yes, but there is a Sec. 4.1 is unclear in that if the transformation A_{x, y} can be obtained using a closed form from the correspondence matrix, why do the authors propose to then again learn the transformation in Sec. 4.2.

Clarity: Yes, it is clear.

Relation to Prior Work: Yes, this is discussed in Sec. 2, 3. and L237-247.

Reproducibility: No

Additional Feedback: ### POST REBUTTAL ### My initial rating was due to my misunderstanding of the text. The authors clarified it in the rebuttal so I will increase my score from marginally below to marginally above acceptance threshold.

[Author Response · NeurIPS 2020]

We thank the reviewers for their valuable and detailed comments. Before addressing individual concerns, we highlight
that our paper is the first functional map-based method that computes correspondences directly on point clouds and
uses a learned basis, made possible through our linearly-invariant embedding formulation. We believe that this will
encourage further work as highlighted by the reviewers and pointed out in the following.

We will release the source code to reproduce all the results and we will include all the details about our implementation
and the hyperparameters in the supplementary material. We will also add all the requested additional analysis and
discussions. If we have enough space, we will move a compact version of Fig. 1 from the supplementary materials to
the main manuscript.

**Necessity of learning the transformation** $A_{\mathcal{XY}}$ **(R4):** The goal of our method is to predict unknown correspondences
between a pair of shapes at test time. The closed-form expression for $A_{\mathcal{XY}}$ assumes the knowledge of ground truth
correspondences, which is only available during training. At test time, we use the learned transformation matrix to
estimate the correspondences. We will clarify this in the final version.

**Impact of the transformation matrix (R1):** As suggested by **R1**, the inset table here
shows the results with a 60 dimensional Universal Embedding (Uni). We consider
that an improvement of 20% from the baseline is a promising starting point for the
first approach in a new direction. Furthermore, in the partial experiment we show that
our basis is very robust, while the Uni suffers from this kind of changes.

|  | noNoise | Noise |
|---|---|---|
| **our** | **5.4e-2** | **6.6e-2** |
| Uni20 | 7.5e-2 | 8.5e-2 |
| Uni60 | 6.9e-2 | 8.1e-2 |

**Alternative losses for Eq. 3:** The loss proposed in Eq. 3 is a simple one that is supported by existing work. We tested
extensively all of the alternative losses mentioned on lines 237-247 of the main manuscript and will be happy to provide
the results and the properties of the resulting bases as an ablation study.

**Outliers in the experiments (R1):** Our method is very robust. In the inset table, we show
quantitative results on **100 pairs** of the test set with 30% outlier points, compared to the best
baselines. We also show a qualitative example on one shape pair in the figure below. We will
be happy to include experiments with different levels and density of outliers in the final version.
Note also that we considered different kind of noise in Figures 2 and 3 of the main manuscript. The training set does
not contain this type of data, highlighting the robustness of our method.

|  | Outlier |
|---|---|
| **our** | **2.7e-1** |
| Uni60 | 3.5e-1 |
| GFM | 3.5e-1 |

**Limitations (R2):** The main limitations of our method are its global and supervised
nature. We believe that extending our method to use a multiscale feature extractor
and unsupervised losses are both remarkable future directions.

**Baselines (R2):** While we do not compare to Deep Functional maps [30] (which
relies on having a mesh as input for its feature extractor), we compare to the most
recent state-of-the-art method GFM [12] that outperforms [30].

*Qualitative result with outliers*

**Stability of the basis (R2):** We provide visualizations of the estimated basis (resp. descriptors) on some real scans
from the FAUST dataset in Fig. 4 and 5 (Fig. 6, 7, 8) of the supplementary. These scans present missing parts, holes and
partiality and are represented by different samplings. We will be happy to include additional visualizations, showing the
stability and robustness to different sampling.

**Additional experiments (R3):** We will be happy to give more details about the adopted evaluation as suggested by **R3**
(Figure 2 and 3) and on the results obtained through our pipeline adding more in-depth analysis, including adding more
noise/outlier results and using different basis sizes as mentioned above.

**Descriptors from basis (R2):** Although this is not our main goal, the use of our learned embedding as a basis for
descriptor computation as proposed by **R2** is an excellent idea. We will be happy to add illustrations to highlight
properties of the basis. Note that the additional request for the basis to be orthonormal could be well-suited in this case.

**Two stage training (R4):** Using two stage training helps to regularize the network. Importantly our embedding network
loss is designed to promote the existence of *some* linear transformation across different embeddings, which is exploited
by our transformation network. We will clarify this.

**Notation and references (R1, R2, R4):** We refer to *linear invariance* since the loss we use to train the embedding is
invariant to linear transformations. We will clarify this and will move the PointNet sentence mentioned by **R2** to the
main body. We will also clarify the notation (including line $231 - A_{\mathcal{XY}}$ should be marked gt), fix the typos and include
all the references suggested by the reviewers.

[Meta-Review · NeurIPS 2020]

Pros: - Use functional maps framework to motivate an a data-driven pipeline - First fully-differentiable functional maps pipeline, in which both the probe functions and the functional basis are learned from data - Well motivated two stage architecture that seems to bring clear benefits - Very well written Cons: - Benchmarks are not standard: missing FAUST official benchmark and partiality is shown on their own test set instead of more common projections or "cuts and holes" - Ablations missing: they don’t explain why they use small number of points (1K), nor why the number of basis is quite small (20) (which could limit applicability to real scans) R1 and R4 were satisfied with the answers provided by the authors and decided to increase their scores to marginally above acceptance threshold. R2 maintains his/her score as marginally above acceptance threshold. He/She considers that the direction proposed in this work is good and the results are promising, but considers that is not yet ready for publication (see the updated review for a detailed description). The AC considers that the paper has merit, is very elegant and shows very promising results. The AC recommends acceptance and encourages the authors to incorporate the suggestions by R2, most importantly the missing ablations (explanations) regarding the number of points and basis functions used.